# Social Undermining and Employee Creativity: The Mediating Role of Interpersonal Distrust and Knowledge Hiding

**DOI:** 10.3390/bs12020025

**Published:** 2022-01-26

**Authors:** Muhammad Arsalan Khan, Omer Farooq Malik, Asif Shahzad

**Affiliations:** Department of Management Sciences, COMSATS University Islamabad, Park Road, Tarlai Kalan, Islamabad 45550, Pakistan; omer_farooq@comsats.edu.pk (O.F.M.); dr.asifshahzad18@gmail.com (A.S.)

**Keywords:** social undermining, interpersonal distrust, knowledge hiding, employee creativity, social exchange theory

## Abstract

This study aims to examine how social undermining restrains employee creativity. Specifically, an attempt is made to investigate the serial mediating role of interpersonal distrust and knowledge hiding in the relationship between social undermining and employee creativity. This study used purposive sampling to draw 309 employees from the advertising agencies of Pakistan. We used a time-lagged research design to collect the data on the measures at three different points in time. A self-administered questionnaire was used for the collection of data. We followed variance-based structural equation modeling (SEM) to conduct the data analysis in SmartPLS. Our study results indicated a significant negative association between social undermining and employee creativity, while serial mediation analysis showed that interpersonal distrust and knowledge hiding partially mediated the above linkage. This study’s findings contribute to the literature on employee creativity by identifying and testing social undermining as an interpersonal inhibitor factor that impairs employee creativity, and this relationship is serially mediated by interpersonal distrust and knowledge hiding. This study offers valuable insights for the managers of advertising agencies.

## 1. Introduction

In today’s competitive business environment, organizations are compelled to develop capabilities for change and creativity in order to survive in a continuously changing business environment [1]. Additionally, the ever-increasing quality expectations and rapid digitalization have focused organizations on fostering employee creativity in order to achieve sustainability and differentiation. Dynamic organizations are urging their employees to be creative, as employee creativity is considered a foundation toward organizational growth. Maintaining and offering a high-quality and stress-free work environment is beneficial for fostering creative work environments. Otherwise, employees working in highly demanding and competitive conditions may experience emotionally draining states, which stimulate workplace stressors that hamper employee creativity [2].

Earlier studies have predominantly explored the factors that foster creativity in employees [1,3], and little is known about the factors that inhibit employee creativity [4,5]. In this contribution, we intend to focus on the influence of the negative aspects of social relationships that may hamper employee creativity, particularly social undermining. Social undermining is distinguished as a hindrance stressor that has several deleterious outcomes for individuals and organizations [6]. Employees who experience social undermining represent a lack of socio-emotional resources in the workplace, including belittling coworkers’ ideas, backstabbing, and spreading false rumors about coworkers. Limited empirical studies have observed the relationships between social undermining and creativity [7]. At the same time, the process that explains how experiencing social undermining in the workplace impedes employee creativity is not clearly understood. The present study attempts to explicate the underlying process by drawing arguments from social exchange theory and the norms of reciprocity [8,9]. Specifically, we propose inspecting interpersonal distrust and knowledge hiding as serial mediating variables in between the relationship of social undermining and employee creativity. Interpersonal distrust is considered a destructive factor that harms the foundation of social relationships in the workplace [10]. The presence of interpersonal distrust among social relationships may not only threaten cooperation with others but even equally endanger one’s own creativity at work [11]. Aside from that, knowledge hiding is a deliberate effort to withhold knowledge and information from other organizational members and even prevent it from being shared upon request [12]. Employees can exhibit interpersonal distrust and knowledge hiding when they perceive themselves as a victim of workplace mistreatment in the form of social undermining, which ultimately negatively impacts their own creativity [13].

The present study examines social undermining and its relative influence on employees working in the advertising agencies of Pakistan. Advertising agencies develop adverts and other related creative promotional campaigns for their clients, who are usually brands. There is no second opinion that creativity is the soul of advertisement, because the generation of new ideas, uniqueness, and novelty is a prerequisite for advertising effectiveness [14]. Therefore, to cope with the challenges of uniqueness and creativity, advertising agencies mainly depend upon the creative abilities of their employees in order to achieve competitiveness. An employee’s creative ability often depends upon the combination of different perceptions and procedures to which workers are exposed via social interactions at work. The working environments in advertising agencies are highly demanding and competitive [14]. When working within tight deadlines, competition elicits fear, and discretionary judgment can induce negative social interactions such as social undermining at the workplace, which inhibits employee creativity via interpersonal distrust and knowledge hiding.

The theoretical perspective and empirical findings of this research study offer significant contributions to the literature on social undermining and creativity, respectively. First, we aim to answer the research calls by offering an enhanced understanding regarding the negative relationship between social undermining and employee creativity [6]. Secondly, this study contributes to the sparse body of knowledge by advancing the understanding of the underlying mechanisms through which social undermining influences employee creativity. We therefore empirically confirm interpersonal distrust and knowledge hiding as serial mediators in the relationship between social undermining and employee creativity. Lastly, this study provides empirical evidence of its injurious outcomes in the significant context of advertising agencies in Pakistan.

## 2. Literature Review and Hypotheses Development

### 2.1. Social Undermining and Employee Creativity

Creativity is achieved through different thinking and a combination of existing earlier, unrelated information, knowledge, approaches, or processes that generate something novel [15]. Creativity is a social interaction process that builds on the foundation of motivation, expertise, and creative thinking skills [16]. The earlier literature revealed several workplace factors that decrease an employee’s creativity, such as abusive supervision [5], social undermining [7], and bullying [17]. Social undermining is a workplace stressor that manifests in different forms which may affect employees’ own feelings about themselves, confining self-control and their abilities to perform their work task creativity. The presence of social undermining may threaten an employee’s success and reputation within the workplace [18]. It entails indirect actions like passing offensive remarks about others, completely ignoring somebody, or deliberately degrading others’ ideas [19]. In line with social exchange theory [8], social undermining is categorized as a harmful interpersonal behavior which evolves negative social exchange relationships in the workplace [20]. Perpetrators may display social undermining behaviors by withholding needed information and deliberately working slowly with an intent to harm the target [18]. Employees who are the target of social undermining are more likely to experience social estrangement in the workplace, which diminishes the level of self-confidence and hampers their attention to existing knowledge, concepts, and required skills that are needed to generate creative concepts [11,21]. Thus, it is reasonable to assume that socially undermining behaviors may diminish creativity. Earlier empirical studies have also found the negative relationship between social undermining and creativity [7]. Based on the above analysis, the following is hypothesized:

**Hypothesis** **1** **(H1).***Social undermining is negatively related to employee creativity*.

### 2.2. Mediating Role of Interpersonal Distrust

Our study assumes that interpersonal distrust transmits the effect of social undermining on employee creativity. This assumption is consistent with the social exchange perspective, where targeted actions contribute to an employee’s victimizations [22]. When a target exhibits negative behaviors in the workplace, his or her actions will be questionable in a mutual social exchange interaction because other members are perceiving him or her as an untrustworthy person in the exchange process [23]. In the workplace setting, individuals are believed to mutually put their efforts toward accomplishing organizational goals. Social undermining infringes this belief, because the perpetrator commonly shows a lack of “socio-emotional” resources in the exchange process, which evolves into distrust among organizational associates [24]. Grovier [10] conceptualized interpersonal distrust as a “lack of confidence in the other, a concern that the other may act to harm one, and that the other does not care about one’s welfare, intends to act harmfully, or is hostile” (p. 240). Interpersonal distrust may decrease the individual’s organizational citizenship behavior and increase the propensity for negative outcomes [25]. Scott et al. [22] found the mediating role of interpersonal distrust in the relationship between target mistreatment and workplace exclusion. Similarly, Mayer and Mussweiler, [26] found a significant association between distrust and creativity. Based on the above evidence, we concluded the following:

**Hypothesis** **2** **(H2).***Interpersonal distrust mediates the relationship between social undermining and employee creativity*.

### 2.3. Mediating Role of Knowledge Hiding

We assume that the relationship between social undermining and creativity is explained by knowledge hiding. According to social exchange theory [27], perpetrators frequently hold contempt for the norm of reciprocity, which is assumed to develop an unconducive relational exchange at the workplace. When employees perceive that they are the victim of maltreatment, they may be compelled to respond to these practices by displaying covert behaviors in exchange, like knowledge hiding [13]. Knowledge hiding is a negative behavior that is embedded in almost all workspaces and has several deleterious outcomes [28]. The existence of knowledge hiding behaviors may affect the flow of information and promote the culture that leads to affecting organizational outputs [11] and developing low-quality social exchange processes [29]. To be precise, employees barring themselves from the network of knowledge exchange will hinder their own abilities to access the collective exchange of knowledge [30]. It is more likely that employees in this degraded social exchange may invest their resources into attaining knowledge that has already been obtained by other organizational members who have deliberately hidden it [12,31,32]. Knowledge hiding may simply confine individuals from acquiring the existing information and knowledge that they need to develop novel ideas [33]. Based on the existing similar studies that anticipated the relationship between social undermining and knowledge hiding [7] and the decremental outcomes of knowledge hiding on employee creativity [5,13], we hypothesized an indirect effect of knowledge hiding in the relationship between social undermining and employee creativity. Individuals who are highly vulnerable to social undermining in the workplace may reciprocate this mistreatment by withholding pivotal knowledge from their organizational associates. Prior studies, such as that of Černe et al. [11], found a significant relationship between knowledge hiding and employee creativity. Thus, based on the above evidence, we hypothesized the following:

**Hypothesis** **3** **(H3).***Knowledge hiding mediates the relationship between social undermining and employee creativity*.

### 2.4. Serial Mediating Role of Interpersonal Distrust and Knowledge Hiding

Social undermining is theorized as an unpleasant behavior that perpetrators may exhibit against the target individuals in the workplace [20]. It adversely affects the target individuals by hampering their abilities to develop and maintain constructive social interaction at work. An individual being the target of social undermining may arouse the feeling of interpersonal distrust toward other individuals [19] because individuals are involved in the social exchange process, where their actions reciprocate each other [8]. According to social exchange theory, interpersonal distrust may stimulate the feeling of uncertainty and generate disbelief among individuals against the unfair other persons [27,34]. Unfortunately, the norm of negative reciprocity also exists in the work context. When one individual may receive mistreatment from other individuals like knowledge hiding, they may develop a feeling of distrust for their opponent [11]. Distrust against one individual in the dyad relationship may elicit the other individuals to reciprocate the same behavior. Individuals usually hide their knowledge from the person whom they distrust because they either retaliate to their act or want to punish them [35]. Knowledge is a valuable resource that a person requires to retain a competitive edge and which may be exchanged evenly in social interactions among other members within the workplace [8]. Nevertheless, interpersonal distrust may serve as the foundation for futile social interaction [8]. Hence, in the knowledge-based economy, individuals may hide their valuable knowledge from other members with an intent to thwart the loss of their knowledge edge or its misuse by others when a knowledge request is present [35]. Past empirical studies have shown that interpersonal distrust is related to knowledge hiding in the workplace [12,35]. It is also evident that knowledge hiding significantly affects the employee’s creative performance at the job [4,5]. Hence, it is plausible to hypothesize that the relationship between social undermining and employee creativity is explained through interpersonal distrust and knowledge hiding. Thus, the following is concluded:

**Hypothesis** **4** **(H4).***Interpersonal distrust and knowledge hiding serially mediate the negative relationship between social undermining and employee creativity*.

## 3. Participants and Procedure

The participants were full-time employees working within the marketing and creative service departments of advertising agencies located in three major cities of Pakistan: Islamabad, Lahore, and Karachi. In this study, a time-lagged research design was utilized where the measurement of the predictor and criterion variables was temporally separated with an interval of 1 month. This design enabled us to reduce the potential concern of common method bias associated with self-administered questionnaires [36]. The purposive sampling technique was used to select a sample of 588 participants, out of which, in the first wave (Time 1), 387 employees filled out questionnaires (response rate of 65.81%) which included items measuring the independent variable of social undermining. One month later, in the second wave (Time 2), the same respondents (who filled out the questionnaires earlier in Time 1) were asked to complete the questionnaire on mediating variables that included interpersonal distrust and knowledge hiding (341 filled it out, with a response rate of 88.11%). Approximately another month later, in the third wave (Time 3), those respondents who participated in the first and second waves were again requested to fill out the questionnaire on the dependent variable of employee creativity (309 complete responses were received with a response rate of 90.61%). A unique identification number was assigned to each respondent in every wave to cross-match the responses collected in each wave. A total of 309 questionnaires were found to be suitable for the analysis. The sample contained 56.3% male and 43.7% female respondents with an average age of 30 years. The results regarding respondents’ qualifications concluded that most of the respondents had bachelor’s degrees (53.4%), followed by 45.6% of the employees holding master’s degrees. The results showed that the majority of the respondent group had work experience under the range of 1–5 years (44.3%), and 37.5% of the respondent group had work experience in the range of 6–10 years.

### 3.1. Measures

This study had used reliable and valid measures to assess the constructs under study. All the scales were measured on a 5-point Likert scale.

### 3.2. Social Undermining

We used the 13-item subscale to measure the undermining perpetrated by co-workers [20]. An original scale consists of 26 items in which each subscale is measured with 13 items. An example item is “How often has the coworker closest to you intentionally insulted you?” Composite reliability was 0.956.

### 3.3. Interpersonal Distrust

A 5-item scale was used to assess the interpersonal distrust [37]. A sample item is “The more I know about this person, the more cautious I become”. Composite reliability was 0.940.

### 3.4. Knowledge Hiding

We adapted the 3 items from the original scale of knowledge withholding to measure knowledge-hiding behavior [28]. A self-reported mechanism was used for knowledge-hiding items, and therefore the original scale was adjusted minimally, such as changing from “Withhold helpful information or knowledge from others” to “I withhold helpful information or knowledge from others”. Composite reliability was 0.914.

### 3.5. Employee Creativity

We used the Zhou and George [38] 13-item scale to assess employee creativity. The original scale was adjusted for the subject “I” included at the start of each item, such as changing “Is not afraid to take risks” to “I am not afraid to take risks”. Composite reliability was 0.949.

### 3.6. Control Variables

The present study controlled the effects of age, gender, and work experience. The literature available on knowledge management and creativity specifies that these demographic factors influence employee creativity [39,40].

## 4. Evaluation of the Measurement Model

We used a variance-based SEM approach to estimate both the measurement and the structural model in SmartPLS. We evaluated the measurement model before the estimation of the structural paths. We performed a confirmatory factor analysis (CFA) to check the reliability and construct validity of the measurement scales. All the alpha coefficient, composite reliability, and average variance extracted (AVE) values exceeded the threshold of 0.7, 0.7, and 0.5, respectively [41]. For the determination of convergent validity, factor loading of the scale indicators on their corresponding factors was evaluated. All the factor loadings of the scale indicators crossed the recommended value of >0.7 [41], showing the strong correlation with their respective constructs. The factor loading of only one item, EC10 (i.e., 0.667), was below the threshold level of >0.7. This item was retained in the final model, since loadings >0.5 and <0.7 are considered acceptable if the composite reliability and AVE values of the respective construct meet the recommended criteria (CR > 0.7 and AVE > 0.5) [42]. To determine the discriminant validity, we used the Fornell–Larcker criterion, where the square root of the AVE of each construct should be higher than the construct’s respective correlation with all the other latent variables [41]. Table 1 and Table 2 depict the acceptable results of the measurement properties of all the scales.

## 5. Hypothesis Testing

Following the process recommended in the PLS-SEM literature, a bootstrapping procedure was followed with 5000 resamples to examine the hypothesized relationships among the constructs proposed in this study. The hypotheses were tested by determining the *t*-values (*t* > 1.96) along with their respective *p*-values (*p* < 0.05) and path coefficients (β), with a standardized value (β > 0.10) being significant. To evaluate the paths, we examined the direct and indirect effects (see Table 3 & Figure 1). The results revealed that the direct effect of social undermining on employee creativity was negative and significant (β = −0.253, *t* = 3.573, and *p* < 0.000, supporting Hypothesis 1).

Examination of the indirect effects for Hypotheses 2–4 revealed that the effect of social undermining on employee creativity through interpersonal distrust was significant (β = −0.128, *t* = 2.945, and *p* < 0.003, supporting Hypothesis 2). In addition, the indirect effect through knowledge hiding was significant (β = −0.066, *t* = 2.869, and *p* < 0.004, supporting Hypothesis 3). For Hypothesis 4, the indirect effect of social undermining on employee creativity through a sequential effect of interpersonal distrust and knowledge hiding was significant (β = −0.046, *t* = 2.745, and *p* < 0.006, supporting Hypothesis 4). This showed that the relationship between social undermining and employee creativity was partially mediated by the serial mediators’ interpersonal distrust (M1) and knowledge hiding (M2).

## 6. Discussion

Research on employee creativity has mainly focused on the facilitating indicators that promote creativity. However, limited research attention has been devoted to identifying the hindrance factors that hamper employee creativity. This study filled this gap by investigating one such inhibiting factor (i.e., social undermining) that affects employee creativity. Based on social exchange theory [8,27], we proposed and examined the mediating role of interpersonal distrust and knowledge hiding in the relationship between social undermining and employee creativity. Our results showed that social undermining is significantly related to employee creativity. This finding is consistent with the perspective of social exchange, suggesting that the perpetration of negative behaviors will be paid back with similar antagonistic behaviors from those on the receiving end [43]. The findings revealed that the occurrence of social undermining in the workplace can impair the cognitive ability to generate creative ideas. This result supports the past empirical evidence suggesting that the exposure to social undermining is negatively related to creativity [7]. Additionally, our data analysis confirmed the proposed mediation hypotheses. The prime finding of this study is that the relationship between social undermining and employee creativity was serially mediated by interpersonal distrust and knowledge hiding, though the literature on creativity has reported the findings of simple mediation analysis [13,26]. However, the serial mediation mechanism has been rarely explored in the studies of creativity. By doing so, this study extends the empirical literature on creativity by testing the underlying mechanism between social undermining and employees’ creativity [11]. Our findings support the norm of reciprocity [9], demonstrating that exposure to social undermining is more likely to prompt a mindset of distrust in the interactional process, which may lead to encouraging negative behaviors (i.e., knowledge hiding). In turn, knowledge hiding from organizational associates would adversely affect the individual’s own cognitive abilities, which impairs his or her ability to provide creative ideas or solutions for concern issues. These findings are similar to those of existing studies to some extent, demonstrating that interpersonal distrust is related to knowledge hiding [35], and in turn, knowledge hiding predicts employee creativity [4,5].

Based on these findings, our study provides managerial implications for advertising organizations. It is noted that social undermining is a stressful factor in the workplace that obstructs employees’ abilities to deliver creative performances. Our findings inform the managers of advertising organizations to take proactive measures to reduce the occurrence of social undermining incidents in the workplace. Managers may conduct training programs that enable employees to recognize potential stressors and separate them from their related relationships [44]. Managers must develop a work environment that fosters creative behaviors and performance. Moreover, the managers should know that recruiting candidates only on educational grounds does not promise creativity. For the enhancement of creative behaviors among employees, it is the main responsibility of the manager to generate cooperative environments that encourage creativity [44,45]. Our findings also showed that knowledge hiding and interpersonal distrust explain the negative effect of social undermining on employee creativity. This finding implies that managers should promote knowledge sharing within the workplace by developing a trust loop in social interactions. Managers should put their efforts toward promoting positive reciprocal relationships and relational interactions of employees within the organizations [5]. Moreover, if it is ensured to employees that they can gain mutual benefits from other organizational members by the exchange of their knowledge information, they would probably consider the information exchange process beneficial, which ultimately increases employees’ creativity.

## 7. Limitations and Future Research Directions

This study has a few limitations which provide directions for further investigation. First, we used a self-administered questionnaire for the data collection process, which is subject to common method variance [36]. However, we believe that common method variance did not affect the study findings because the data were collected at three different points in time. The design allowed us to minimize the potential biases at the time of the survey. We invite future studies to use other rating data (e.g., supervisors and co-workers) to measure employee creativity. Second, although the present study used a time-lagged research design which only reduced common method variance, this design could not rule out the possible reverse or reciprocal associations among the study variables. Therefore, we encourage future research to apply a longitudinal research design to address the directionality issues among creativity and its predictors. Third, for this study, we slightly modified the scales of knowledge hiding and employee creativity. Although these scales were adapted from validated scales, future research should work on the same scales to confirm the findings of this study. Fourth, we tested our research framework in the context of advertising organizations, and therefore the findings of our study should be applied to other work contexts with caution. We call for further investigations to test our research framework in other industries to increase the generalizability of our study results.

## 8. Conclusions

Our study adds to the body of knowledge by validating social undermining as an interpersonal inhibiting factor of employee creativity. Additionally, this study identifies the underlying mechanism through which employee creativity is hampered. We expect that our study will serve as a catalyst for further research on how organizations can manage the harmful impacts of various unpleasant working conditions on employee creativity.

## Figures and Tables

**Figure 1 behavsci-12-00025-f001:**
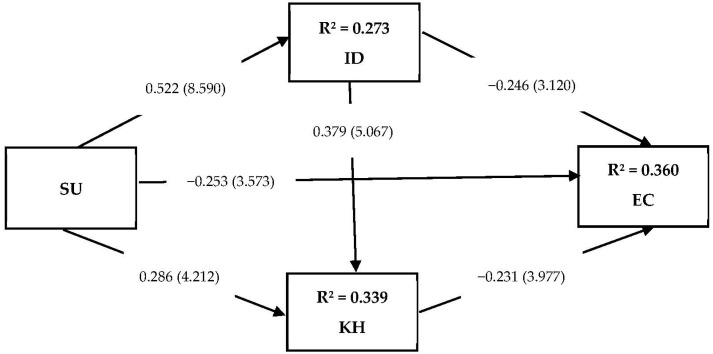
A serial mediation model, with *t*-values in brackets.

**Table 1 behavsci-12-00025-t001:** Psychometric properties of the scales.

Construct	Indicator	Loading	Cronbach’s Alpha (α)	AVE
Social Undermining	SU1	0.794	0.950	0.627
	SU2	0.789		
	SU3	0.800		
	SU4	0.817		
	SU5	0.802		
	SU6	0.798		
	SU7	0.815		
	SU8	0.800		
	SU9	0.801		
	SU10	0.798		
	SU11	0.800		
	SU12	0.754		
	SU13	0.724		
Interpersonal Distrust	ID1	0.869	0.921	0.760
	ID2	0.877		
	ID3	0.865		
	ID4	0.874		
	ID5	0.873		
Knowledge Hiding	KH1	0.914	0.860	0.781
	KH2	0.876		
	KH3	0.860		
Employee Creativity	EC1	0.758	0.941	0.588
	EC2	0.708		
	EC3	0.764		
	EC4	0.772		
	EC5	0.810		
	EC6	0.791		
	EC7	0.793		
	EC8	0.807		
	EC9	0.786		
	EC10	0.667		
	EC11	0.756		
	EC12	0.796		
	EC13	0.746		

**Table 2 behavsci-12-00025-t002:** Descriptive statistics and discriminant validity.

Construct	Mean	Std	EC	ID	KH	SU
Employee Creativity	2.379	0.665	**0.767**			
Interpersonal Distrust	3.346	0.916	−0.500	**0.872**		
Knowledge Hiding	3.205	0.991	−0.483	0.529	**0.884**	
Social Undermining	3.456	0.867	−0.494	0.522	0.484	**0.792**

Note: Bold diagonal values are square roots of AVE.

**Table 3 behavsci-12-00025-t003:** Direct and indirect effects of social undermining on employee creativity.

Path	Coefficient	*p*-Value	*t*-Value	BC 95% CI	Result
SU → EC	−0.253	0.000	3.573	[−0.385, −0.113]	H1, Accepted
SU → ID → EC	−0.128	0.003	2.945	[−0.219, −0.051]	H2, Accepted
SU → KH → EC	−0.066	0.004	2.869	[−0.119, −0.029]	H3, Accepted
SU → ID → KH → EC	−0.046	0.006	2.745	[−0.083, −0.019]	H4, Accepted

Note: BC = bias corrected, CI = confidence interval, SU = social undermining, ID = interpersonal distrust, KH = knowledge hiding, and EC = employee creativity.

## Data Availability

The data used to support the findings of this study are available on request from corresponding author.

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
