# Peer review of "Social Undermining and Employee Creativity: The Mediating Role of Interpersonal Distrust and Knowledge Hiding"

_behavsci, 2022, doi:10.3390/bs12020025_

Round 1

Reviewer 1 Report

The present research study aimed to investigate the impact of social undermining on employee creativity. Additionally, the mediating role of interpersonal distrust and knowledge concealment was examined. In order to eliminate common method error, the study was conducted according to a time-wave model. The authors report that social undermining has a significant negative impact on employee creativity. The mediation analysis performed confirmed the interaction of interpersonal distrust and knowledge concealment between social undermining and creativity relationship. 

However, the study has limitations. The main one is the use of a questionnaire of the authors' own design, about which there are no reports of psychometric properties. Additionally, the Authors modified other questionnaires - which should be clearly highlighted in the limitations of this study. In my opinion, the authors should have described the limitations of this study in more detail. The authors also did not include information whether the study was approved by the research ethics committee. This information should be supplemented.

The results presented here are important because to date research on employee creativity has focused mainly on enabling indicators, but little attention has been paid to identifying inhibiting factors. This paper fills this gap by examining one such inhibiting factor - social undermining. The authors have also developed a theoretical framework for the relationships studied. Based on social exchange theory, they propose the mediating role of interpersonal distrust and knowledge concealment in the relationship between social undermining and employee creativity.

The authors also describe the possible application of the results and the actions that should be taken in the managerial field in order not to inhibit employee creativity.

Author Response

We are thankful to the respected reviewer for reading the research study keenly and highlighting the relevant questions. We have tried to accommodate the changes accordingly. 

File attached

Reviewer 2 Report

Abstract

The abstract generally explains the study and its rationale clearly. The recommendations could use some honing. It would be desirable for the final section in the paper to mention which aspects of the study would cross-apply to professional environments and why the results can be extrapolated beyond educational coursework. The Methodology component of the abstract should clarify the types of data analytics used.

Methodology

This is one of the most critical parts of the paper that I found lacking detail.

The methods should be adequately described to show how the research was conducted to improve clarity and transparency.

Instruments

The section must devote details to the description of the instrument.

Expand the development process of the instrument.

 Data analysis 

The quality of statistical reporting and data presentation in the paper is inadequate.

Conclusion.

The finding should be related to results from previous literature. The conclusion section should highlight the unique contributions of the paper and the limitations of the research. Discussions on what should be done in the future are useful. The discussion and conclusion should make it clear how the research findings contribute to new knowledge.

Plagiarism check results:

/* Similarity check with iThenticate revealed a similarity index of 19%, which is considered NOT appropriate. A maximum of around 60 quoted words is accepted per paper. There are papers with over 60 words. No previously copyrighted material was used.  

Author Response

We are thankful to the respected reviewer for reading the research study keenly and highlighting the relevant questions. We have tried to accommodate the changes accordingly. 

File Attached

Reviewer 3 Report

Title - the title should not include the name of the statistical technique used. Abstract The abstract is too long. Ideally, it should contain only basic information: Background, Material and Methods, Results, Conclusions. 1. Introduction "In today’s competitive business environment, organizations are compelled to develop capabilities for change and creativity inorder to survive in a continuously changing business environment [1]." - Avoid repetitions. In this section, it is worth referring to the theory of organizational justice and counter-productive work behavior. 2. Literature Review The authors should emphasize why social undermining is so important to creativity. The hypotheses were correctly constructed. 3. Methods Please provide the average age of the respondents. 3.1. Measures The research tools are presented correctly. Usually, when describing the tools, the reliability factor is given: Alpha (Cronbach) or Omega (McDonald). 4. Results „4.2. Hypotheses Testing PLS bootstrapping was used to compile the results to test the four hypotheses proposed in this study.” - please present the number of sampling 5. Discussion The discussion contains the most important information in relation to other studies. This part does not need to be improved. It is important that the Authors provide practical advice for managers.

Author Response

(The authors gave the same response as above.)

Round 2

Reviewer 2 Report

Overall, the revised manuscript adds and improves upon the first. I was generally satisfied with the authors' response to my questions and comments.

Author Response

Thank You for highlighting the relevant point. We have proof read the paper completely and rectified the mistakes related to spelling and grammar check.